



# Groundings of Drifters in the Wadden Sea Inform the Transport of Floating Macroplastic

Marc Emanuel Schneiter[1], Rolf Hut[2], and Erik van Sebille[1]

[1]Institute for Marine and Atmospheric Research (IMAU), Utrecht University, Utrecht (NL)
[2]Department of Water Management, Delft University of Technology, Delft (NL)

**Correspondence:** Marc Emanuel Schneiter (m.e.schneiter@uu.nl), Rolf Hut (r.w.hut@tudelft.nl), and Erik van Sebille
(e.vansebille@uu.nl)

**Abstract.** We study floating marine macro plastic pollution with a dedicated campaign where we released and tracked 24
shallow surface drifters in the Dutch Wadden Sea, an intertidal region where water dynamics are driven by semi-diurnal tides
and wind. The campaign captured water surface transport patterns, and around 150 observations of the drifters interacting
with the coast and mudflats, where groundings of varying durations occurred. We identify these transitions in the recorded
trajectories from both manual inspection and by formulating and evaluating a statistical approach for automatic detection. We
then characterize the transitions by relating them to the coinciding wind conditions and tides, and with the duration of the
groundings. The results of this analysis generally agree with the intuitions that wind towards the coast increases the frequency
of interactions and that the drifters are rarely re-floated during falling tides. Furthermore, we find that groundings tend to
be shorter duration when they occur shorter before the next high tide. We present multiple visualizations that characterize
the groundings in more detail. The results provide a statistical characterization of coastal interactions that can be integrated
in models and used for the planning of beach-cleanups. They may be improved by considering the specific environmental
conditions of the campaign, and by analyzing the influence of the weight and size of the drifters, which are both larger than
typical floating macroplastic.

## 1 Introduction

The Wadden Sea is a shallow body of water bordering the north coast of the Netherlands and west coast of Germany and Den-
mark, with numerous channels situated between the Frisian Islands that connect it to the North Sea. It holds unique ecosystems
and geological features, which is why it has been listed as an UNESCO world heritage site in 2009 (see Common-Wadden-Sea-
Secretariat; Colina Alonso et al., 2021; Kraan et al., 2010). As a water system, it consists of coastal wetlands and tidal basins
that border the shelf sea across a range of barrier islands and that have multiple inflows from rivers. Its dynamics result from
interactions between the semidiurnal tides of the region and residual currents of the North Sea, from wind stress that gener-
ates waves and surface currents, from river runoffs forming freshwater plumes, and from the bathymetry and shape of the coast



(Deyle et al., 2024; Fajardo-Urbina et al., 2023; Ricker et al., 2021; Meyerjürgens et al., 2019; Maas, 1997; Gräwe et al., 2016).

The prevailing stresses on marine ecosystems through climate change, marine pollution and other unsustainable interventions are also concerns for the Wadden Sea (Eriksson et al., 2010; Kabat et al., 2012). A major contributor to the pollution is plastic debris (Neumann et al., 2014), where areas close to mainland are accumulating regions for floating macroplastic, buoyant pieces of plastic larger than 5cm (Kaandorp et al., 2023). Marine plastic enters the seas as discharge from rivers, mismanaged waste from ships or fisheries, and littering at the coast (Schöneich-Argent and Freund, 2020; González-Fernández et al., 2021).

The increase in use of plastic items over the last decades coincides with a growing contamination of open waters. This poses an environmental threat, both as an obstacle for marine life, as well as by the risks arising from an increase in microscopic pollution after the fragmentation of plastic waste into micro- and nano-plastics (Dris et al., 2020). Litter at beaches also harms the touristic value of regions at the coast. Understanding the transport of floating macroplastic thus contributes to the study and mitigation of these issues.

Hydrodynamic and atmospheric models can be used in Lagrangian particle tracking simulations to model the dispersion of plastic pieces in open waters (Ricker et al., 2021; Critchell et al., 2015; Mørk et al., 2026). Multiple numerical studies have been conducted for the Wadden Sea to examine the transport of water masses, its variability with respect to wind patterns and seasonality, and dispersion (Fajardo-Urbina et al., 2023; Donatelli et al., 2022; Duran-Matute et al., 2014). These make

use of estuarine models (Burchard and Bolding, 2002) and particle tracking (Delandmeter and van Sebille, 2019; Lange and van Sebille, 2017). Accurately representing marine plastic transport in coastal areas also requires a model for littoral-coast transitions, which further increases the complexity of such studies. The reliability of both the simulation outcomes and of the hydrodynamic models can be improved with in-site measurements from stations at the coast or in the water (Meyerjürgens et al., 2019; Morey et al., 2018).

Here, we report on a measurement campaign from November 2023 of 24 floating drifters at the western Wadden Sea. This experimental Lagrangian sampling is the first measurement of its kind in the Dutch Wadden Sea and also provides an additional dataset to the sparsely sampled North Sea area (Deyle et al., 2024; Meyerjürgens et al., 2019; Fajardo-Urbina et al., 2023; Medina-Rubio et al., 2025; Elias et al., 2022). A particular process that is captured in the dataset is the occurrence of grounding

events where drifters temporarily settle down on tidal flats. The focus of the data analysis here lies on these events and groundings at the coast, which are an important component to better understand the displacement of marine debris, and to predict accumulation regions of stranded plastic on the coast (Critchell and Lambrechts, 2016; Meyerjürgens et al., 2019; Pawlowicz, 2021). These forecasts in turn are valuable to beach cleanup efforts, which are mainly organized as volunteering activities. Knowledge about general hot spots and specific beaching patterns can inform decisions on when and where beach cleanups are

most effective. Another motivation to study grounding of macroplastic is that the transitions between water and land take part in the mechanical decomposition of the plastic (Takeda and Isobe, 2024; Kaandorp et al., 2021).



We describe two systematic approaches to manually and to statistically identify grounding events in trajectories, and apply them to the campaign observations. By comparing the two outcomes, we verify the reliability of the grounding statistic and its parameterization. We then classify the groundings with respect to environmental conditions - wind and tidal state - and the grounding duration.



## 2 Methods

### 2.1 Campaign

We released 24 drifters on 14 November 2023, as summarized in Fig. 2. This took place in three locations near the Dutch island Texel, approximately 9km apart, and shortly before and after the low tide at 13:00. For the selection of the release locations and times, we considered the typical sea surface dynamics and local bathymetry, and consulted personnel at the Royal Netherlands Institute for Sea Research (NIOZ). We also estimated the trajectories with numerical simulations and weather forecasts, in particular the tidal cycles on the release day. The intention was for the drifters to be moved out of the Wadden Sea and into the North Sea during a few tidal cycles. We released the drifters as nested triplets in a triangular pattern at two of the three locations, and as two triplets at the third location. From this, trajectories of each triplet can be analyzed as three different pair releases or one triplet release. The release-pattern is self-similar, with the smaller scale of 10m for the nested triplets, and the larger scale of 100m for combinations of triplets. The releases on the smaller scales are only an approximation of a triangular pattern, and the initial drift after the deployment was sometimes influenced by ship maneuvers. One batch of six drifters were re-deployed in a secondary release at multiple locations east from the Dutch island Vlieland on 20 November.

Stokes Drifters were used to collect the measurements. They are shallow and disk-shaped trackers produced by MetOcean (see MetOcean). They have a diameter of 24 centimeters, a height of 4 centimeters and a weight of 1kg. These drifters lie flat in the water, their profile reaches roughly 1.5 centimeters out of the water surface. This makes them sensitive to dynamics in the uppermost centimeters of the water column and to air-drag (Medina-Rubio et al., 2025; Morey et al., 2018). For tracking, the drifters contain a positioning system (GNSS), and for communication, they carry a transmitter for the Iridium satellite system. Furthermore, the Stokes Drifters record the sea surface temperature (SST) and the orientation of the drifter in the water. Power is provided by a set of 10 AA-alkaline batteries. The operational time depends on the sampling configuration, consisting of a sampling interval and an upload interval. The sampling interval determines the time between measurements. We set it to 5 minutes (resulting in time-series with approximately 5.2 minutes intervals) to record with the highest available temporal resolution. The upload interval regulates after how many measurements the collected data is transmitted. Since these transmissions are power intense, we set this value to 12, which resulted in one transmission attempt per hour.

### 2.1.1 Citizen Involvement

The campaign involved citizen contributions both for the funding and the observation and recovery of the drifters. Before the campaign, the University Fund of Utrecht University organized a crowdfunding campaign for the project. During the campaign, citizens were involved because the drifters are spotted once washed ashore. Anticipating this, we designed the sticker shown in Appendix E to convey a brief message to finders. The intention was that finders send an email with a picture of the drifter for documentation, without moving the drifter to another location. In this manner, beaching and refloating processes of the drifters should remain part of the observations. In addition, once a reported drifter had reached the end of its measurement period, we asked the finders to return the drifter via post. We received 35 sightings, and recovered 12 of the 24 released drifters.



### 2.1.2 Data Collection and Processing

We collected the transmitted measurements and manually removed excess measurements from the transport of the drifters by people on land. Some drifters have also been redeployed after a first collection, which we record as independent deployments. The analysis that follows is based on the drifter trajectories of this dataset (van Sebille, 2024).

The raw trajectories are processed to derive a set of homogeneous time-series. In a first step, we remove outliers that appear as abrupt changes of location. It was found that filtering out displacements between consecutive measurements of more than 50km in either the longitudinal or latitudinal direction provides satisfactory results. In the second step we regularized the time-steps of the trajectories by splitting up the trajectories at time-steps larger than a chosen threshold $t_{\max}$, and by averaging the timestamps and positions of consecutive measurements with time-steps smaller than a chosen threshold $t_{\min}$. This step
resulted in a few trajectories with only a small number of positions. We removed these in a third step by requiring a minimum total duration $T_{\min}$ and a minimum number of samples $N_{\min}$ of a time-series. The parameter choice for the analysis here are 20 minutes for $t_{\max}$, 2.5 minutes for $t_{\min}$, 1.5h for $T_{\min}$, and 12 for $N_{\min}$.

We recreate the sea-conditions, the topological surroundings, and the wind-conditions of the drifters from a MATROOS dataset
(see Rijkswaterstaat), a high-resolution regional bathymetry map (see Waddenregister), and the ERA5 reanalysis (see ERA).

## 2.2 Detecting Wetting and Grounding Events to Identify Grounded and Floating Sections

The transport of floating macroplastic in coastal regions can be described with the terminology summarizzed in Table 1, as a sequence of grounding and wetting events that border grounded and floating sections. Fig. 1 shows a drifter-trajectory from the campaign with multiple floating and grounded sections. The characteristic star-shaped pattern is formed by inaccuracies of the
GNSS receivers when the position of the drifter remains constant for multiple measurements. This contrasts with the otherwise directional and smoother features in parts of the trajectory when the drifter is afloat.

**Table 1.** Terminology to describe littoral-coast transitions of drifters or floating macroplastic.

| Term | Type | Definition | Duration |
|---|---|---|---|
| Wetting | Event | Object is flooded and then carried in the water | Instantaneous |
| Grounding | Event | Object is deposited and then resting immobile on land | Instantaneous |
| Drying | Event | Grounding with a subsequent wetting | Instantaneous |
| Beaching | Event | Last recorded grounding | Instantaneous |
| Floating | Section | State after wetting | From wetting time-point to grounding time-point |
| Grounded | Section | State between drying and wetting | From drying time-point to wetting time-point |
| Beached | Section | State after beaching | Undetermined |







**Figure 1.** Typical observations of a drifter during floating and grounded sections: From top to bottom, the plots show the trajectory and two breakouts with a zoom-in on the last 11h and 7h, the change in direction between consecutive measurements (green), the speed (red), the water-level at the drifters position and time (upper blue), and the water depth below the drifter (lower blue) are shown. The color-sequence along the time axis of the line-plots is matched with the way-points of the trajectory. The vertical green and blue lines mark grounding and wetting events. The sampling frequency is 5.2 minutes. Map and coastline generated with (see Met Office, 2010 - 2015)



### 2.2.1 Manual Grounding Identification

We visually inspected and annotated the trajectories with labels for grounding and wetting events, based on the guideline described in Appendix A. For validation, ten trajectories were annotated by two coders. Comparing the two outcomes per
time-step gives an agreement of 94% for one trajectory, and above 97% for the other nine ones. A comparison per label shows that roughly 75% of the labels have a correspondence within five time-steps (30 minutes), 10% agree within up to 19 time-steps (2 hours), and 15% of the annotations do not have a correspondence. The mismatches from the last category are annotations of four shorter grounded sections, where the drying and wetting events are 30-60 minutes apart.

### 2.2.2 Statistical Grounding Identification

The plots in Fig. 1 show that quantities derived from the trajectories display characteristic behaviors in correlation with grounded sections. We subsequently formalize this observation to design a statistic that should be applicable for different GNSS systems and sampling strategies. The GNSS trajectories can be written as time series of positions $p(t) = (\mathrm{lon}(t), \mathrm{lat}(t))$ at sampling times $t = t_1, t_2, ..., t_N$. We measure distances between recorded positions with the geodetic distance $d_g(p_1, p_2) := |p_2 - p_1|$ and locally define angles between displacements $\measuredangle(p_2 - p_1, p_3 - p_1)$ from a projection of the positions to Cartesian
coordinates.

### 2.2.3 Definition of the Grounding Statistic

The speed $v(t)$ as a discrete derivation of the position

$$v(t_i) := \frac{|p(t_{i+1}) - p(t_i)|}{t_{i+1} - t_i}$$

is low when the drifter is immobile. This occurs when a drifter is suspended on land, but it can also be the result of specific
states of the water surface, for example at the turning points of tidal cycles. A related quantity is the displacement $D(t, \Delta T)$ during a chosen time interval $\Delta T$

$$D(t, \Delta T) := |p(t + \Delta T/2) - p(t - \Delta T/2)|$$

where the continuous time $t \pm \Delta T/2$ can be matched to the discrete recorded times $t_i$ by an interpolation scheme. We choose to use nearest neighbor interpolation $p(t) \rightarrow p(t_i)$, $i = \mathrm{argmin}\{|t_i - t|\}$. The distance $D(t, \Delta T)$ remains low when the drifter is
immobile. Since the trajectories of drifters on water surfaces with slow currents usually follow a smooth path at low velocities, this metric is expected to be more suitable than speed for separating situations like the turning points of tides from grounded sections.

The change in direction $\alpha(t)$ over consecutive measurements



$$\alpha(t_i) := \measuredangle(p(t_{i+1}) - p(t_i), p(t_i) - p(t_{i-1}))$$


is low for smooth parts of the GNSS trajectories. We make use of a preprocessing step to standardize the computation of $\alpha(t_i)$, as described in Appendix B. When a drifter is following surface currents, the distances between subsequent measurements are larger than the inaccuracy of the positioning system. This, combined with the continuity of the currents at the sensitivity scale of the drifter, creates a correlation between low values of $\alpha(t)$ and the drifter being afloat. This is in contrast to situations when

the drifter is immobile, where the jitter is the dominant contribution to differences between consecutive measurements. This quasi-random displacement generates star-shaped patterns that coincide with abrupt directional changes and thus large values of $\alpha(t)$ that fluctuate over time. In order to quantify this characteristic behavior, the averaged directional change $A(t, \Delta T)$ during a time interval $\Delta T$ can be computed as

$$A(t, \Delta T) := \frac{1}{|W|} \sum_{t_i \in W} |\alpha(t_i)|$$
$$\text{with } W := W(t, \Delta t) = \{\, t_i \mid t_i - \Delta T/2 \le t \le t_i + \Delta T/2\}$$

Similar to the discussion above about velocity and displacement, the extension of an instantaneous metric to a time window makes it possible to better separate the smoother turns in the trajectory of a floating drifter, where high changes in direction can occur over only a few measurements, from the more constant appearance of large directional changes when the physical position of the drifter does not change.

Both $D$ and $A$ are independent of the time intervals $t_{i+1} - t_i$ between measurements $i \in I$. This choice was motivated by considering trade-offs between accuracy and simplicity of the detection method, and the robustness to sparse and irregular sampling. In principle, one can choose two different time windows $\Delta T_D$ and $\Delta T_A$. The prediction of grounded sections in the time series follows by introducing thresholds $D_{\text{th}}$ and $A_{\text{th}}$. For a time $t_i$, the time-window defines the set of associated entries that are used to test if a threshold is crossed. If this is the case, all entries of this set are considered grounding candidates:

$$\textbf{if } D(t_i, \Delta T_D) < D_{\text{th}} \textbf{ then } \text{grounded}_D(t_i) = \text{True}, \textbf{ for all } t_k \in W(t_i, \Delta T_D)\,,$$

$$\textbf{if } A(t_i, \Delta T_A) < A_{\text{th}} \textbf{ then } \text{grounded}_A(t_i) = \text{True}, \textbf{ for all } t_k \in W(t_i, \Delta T_A)\,.$$


From this, grounding is predicted for an entry where both quantities indicate grounding:

$$\text{grounded}(t_i) := \text{grounded}_D(t_i) \wedge \text{grounded}_A(t_i)\,.$$



### 2.2.4 Parametrization

The statistical method has four parameters summarized in Table 2, which we reduce to three parameters by using one time-window $\Delta T$ for both quantities. Since we rely on a sparse dataset with a limited number of grounded and floating sections, we choose the values of these parameters guided by considerations of characteristics of the measurements and of the physical processes, rather than by using statistical tools.

The time-window $\Delta T$ sets a lower bound for the temporal detection of grounded sections, the duration of which can vary between a few minutes and days. The position sampling frequency implies an average number of entries that fall within each time window, which sets a lower bound for $\Delta T$. Its value should be chosen close to this lower bound, in order to increase the temporal resolution of the grounding detection, and to avoid interference from physical transport processes at higher timescales. In the case of the Wadden Sea, the semidiurnal tidal cycles have a major impact on current patterns. Splitting cycles up into rising-, high-, falling-, and flat-tide phases gives a temporal scale of 3 hours, thus $\Delta T \leq 3\mathrm{h}$.

The apparent displacement of a drifter at rest is caused by the uncertainty of the GNSS system, which sets a lower bound for the choice of $D_{\mathrm{th}}$. This has to be balanced against the slowest observed displacements of a floating drifter during the characteristic time $\Delta T$. This can for example be inferred from the lowest characteristic surface current that the drifter is sensitive to. For the Wadden Sea, slow surface currents may be on the order of 1cm/s and assuming this speed constant during high- or flat-tide leads to a characteristic minimal displacement of $1\mathrm{cm/s} \cdot 3\mathrm{h} \sim 100\mathrm{m} \geq D_{\mathrm{th}}$. Comparing different choices for $D_{\mathrm{th}}$ showed that the standard deviation of the recorded position of a drifter at rest works well for grounding detection. Since the accuracy of a GNSS system can be different in the North and East direction, we compute the standard deviations $\sigma_{\mathrm{N}}, \sigma_{\mathrm{E}}$ in either direction for the trajectory of a drifter at rest, and set $D_{\mathrm{th}} = \max\{\sigma_{\mathrm{N}}, \sigma_{\mathrm{E}}\}$.

Similar arguments can be made for the choice of $A_{\mathrm{th}}$, but the dependencies of sampling accuracy, frequency, regularity, and the movements of the drifter are more complex. An order estimation for a characteristic reorientation of floating drifters comes from assuming that the travel direction is reversed during falling- and rising-tide phases, such that $180°/n(3\mathrm{h}) \leq A_{\mathrm{th}}$, with the average number of samples during 3 hours $n(3\mathrm{h})$. A useful reference to set this threshold can be derived from the assumption that the changes in direction of a GNSS receiver at rest follow an uniform random distribution: For a set of random variables $\{a_i\}_{i=0,1,\dots} \sim \mathrm{Unif}[-180, 180]$ and with the probability $p = \mathbb{P}[\theta_p = (1-p) \cdot 180° \leq |a_0|]$ it follows that $p = \mathbb{P}[\theta_p \leq A|_{\{a_0\}}]$ for one angle. In accordance with the choice of the previous paragraph of one standard deviation for the threshold of the displacement $D_{\mathrm{th}}$, a lower bound for this threshold is thus $p = 0.68 \rightarrow \theta_{0.68} = 57.6° \leq A_{\mathrm{th}}$. On the other hand, the expected value of the metric $\mathbb{E}[A|_{\{a_0, a_1, \dots\}}] = 90°$ is independent of the number of samples in each time-window, providing an upper bound. It was found that setting $A_{\mathrm{th}} = 58°$ resulted in satisfactory results.





**Table 2.** Summary for the parametrization of the grounding statistic.

| Symbol | Name | Selection Criteria | Values Used |
|---|---|---|---|
| $\Delta T_D$ | Time-window displacement $D(t, \Delta T_D)$ | | 180, 90, 45 min |
| $\Delta T_A$ | Time-window averaged directional change $A(t, \Delta T_A)$ | $\Delta T_A = \Delta T_D$ | 180, 90, 45 min |
| $D_{\text{th}}$ | Threshold displacement $D(t, \Delta T_D) < D_{\text{th}}$ | $D_{\text{th}} \approx \sigma_{\text{gnss}}$ | $\sim 6$ m |
| $A_{\text{th}}$ | Threshold averaged directional change $A(t, \Delta T_A) < A_{\text{th}}$ | $58° \leq A_{\text{th}} \leq 90°$ | $58°$ |

### 2.2.5 Evaluation of the Grounding Statistic

Two metrics are used to quantify the agreement of the manually observed groundings with the predictions from the grounding statistic. The first metric makes a comparison for each entry in each time-series. While this approach shows the fraction of correct and incorrect predictions on a time-step basis, it does not indicate the effectiveness of the statistic to identify individual floating and grounded sections. These vary greatly in duration, such that misidentified shorter sections are less apparent. Furthermore, some sections have durations smaller than the chosen time-window and therefore lie outside the scope of the statistic.

For these reasons, a second metric is used that compares observations and predictions on a section basis. When examining a recorded trajectory for groundings, sections where the drifter is afloat should have no indication of grounding for the entire duration, and sections where the drifter is on land should have at least one indication of grounding. For this reason, the second metric counts floating sections with the absence of any grounding prediction as one correct floating prediction, and otherwise counts the number of separate subsections of false grounding indications as wrong floating predictions. A grounded section

counts as one correct grounding prediction if the grounding is indicated at least once, and as one incorrect grounding prediction otherwise. Sections with durations smaller than $\Delta T$ are not taken into account.

The reliability and adaptability of the grounding statistic is evaluated for different re-samplings of the trajectories. The entries were either selected with a uniform stride, or by dropping a fraction of the entries at random. For either method, the

number of entries was repeatedly divided by two, in parallel to the reduction of the time-window for the grounding detection. This results in the average number of entries in the time-windows used for the evaluation being similar for different reductions of the trajectory and/or time-window.

### 2.2.6 Characterization of Groundings

We want to identify environmental conditions and other factors that explain the observed groundings, where we focus on the

situation that a drifter is already in a region where groundings can occur. Since the events happen at the intersection between the water surface and coast, we consider the changes in water level over time in combination with the bathymetry as a proxy for water depth, which in turn informs near coast surface currents, and whether an intertidal zone is flooded or not. Local wind also impacts both the water surface and drifter movements (de Amorim et al., 2025).





Since tides are the main drivers of dynamic changes in the Wadden Sea, we extract the tidal phase over time at a grounding location. This is to determine whether an event happens during high, falling, low or rising tides. Wind is characterized by its speed and its direction with respect to the local orientation of the coast or mudflat. Since determining this orientation appropriately is challenging, we describe our approach and a sensitivity analysis in Appendix C. We separate the local wind direction into four sectors of directly towards shore, diagonally towards shore, diagonally from shore and directly from shore.

The observed groundings occurred on slopes with a large variation of directions, which is why we do not differentiate between the two mirrored wind directions with the same angle to the coast. Furthermore, we determine the duration of grounded sections. Table 3 summarizes the four factors we choose for the analysis.

**Table 3.** Environmental factors and other characteristics chosen to classify groundings. The two illustrations in the right column indicate the local tidal phase and wind direction relative to the shoreline.

| Factor | Range / Bins | Illustration / Remark |
|---|---|---|
| Tidal Phase | [ H, F, L, R ] |  |
| Wind Direction (deg) | [ 0-45, 45-90, 90-135, 135-180 ] |  |
| Wind Speed (m/s) | [ 0-6, 6-12, 12-18 ] |  |
| Section Duration (s) | [<T/4, T/4-T/2, T/2-T, >T ] | T = 12h25min |



# 3 Results

## 3.1 Campaign Overview

The campaign is summarized in Fig. 2 of all trajectories and transition events, Fig. 3 showing the wind-conditions and water-level during the campaign, and Fig. 4 with timelines of the drifting sections, water level and wind conditions. While the water-level timeline shows the dominant component of the semi-diurnal tides, one can also see longer-term trends that persist over multiple days. Wind-conditions evolved on time scales of one to a few days.

We observe that the transitions captured in this campaign are a combination of clustered and isolated events. Two clusterings of drying events occurred in the afternoon of 18 November 2023 and in the morning of 21 November. The clustering on 18 November involves 11 drifters from all three primary release locations that ground at the 1.5km long coast strip of Oost-Vlieland within approximately four hours. The majority of these drifters were then shortly re-floated and accumulated at the dock wall of the port of Vlieland by the morning of 19 November. In contrast, the clustering on 21 November involves nine drifters from the primary and secondary releases, where the groundings occur individually or in small groups at distinctly different locations.

## 3.2 Evaluation of the Grounding Statistic

We use the metrics and resampling-scheme of Sect. 2.2.5, and the parametrization according to Sect. 2.2.4 to evaluate the grounding statistic. Fig. 5 shows confusion matrices where the ground truth values from the manual coding are compared with the predicted labels from the statistic. The number of correct classifications tends to be reduced by a few percent when either the mean number of entries that are available for each prediction are reduced from multiple tens down to only a few, or when evaluating irregularly instead of uniformly sampled trajectories.

## 3.3 Characterization of Groundings

Putting the observed events in relation to the tidal cycle in Fig. 6 shows that wettings occurred predominantly during rising tides and are absent during falling tides, whilst dryings are much more uniformly distributed. Beachings were observed mainly during high tide. While this outcome for beachings and wettings seems intuitively correct, the less expected distribution for drying events can be better understood when also taking the duration of the grounded sections into account: Groundings during low and rising tides are most often short-term, less than one quarter of a tidal cycle. One can observe a shift in the typical duration of drying events from long-term groundings during high and falling tides to shorter events during low and rising tides.

Separating the events by the local wind direction in Fig. 7 shows that the majority of transitions occur during winds towards shore. For dryings and groundings, this asymmetry is more pronounced than it is for wettings. Furthermore, only long-term wettings and short-term dryings occurred during off-shore wind, whereas in contrast all wetting events during on-shore wind



have a duration of at most one tidal cycle. A clear majority of beachings happens during diagonally towards shore wind. As for

265   wind strength, the analysis shows that most events occur during median to strong winds (6-18m/s). This suggests a non-uniform relation with the abundance of wind conditions during the campaign shown in Fig. 3.

Categorizing the events by both the tidal cycle and wind direction in Fig. 8 shows that most observations can be explained by the following statements: Wettings are most often correlated with rising tides, and cross-shore or onshore winds. The oc-

270   currence of dryings is often correlated with diagonally or directly on-shore wind. Beachings were almost exclusively observed under winds diagonally towards shore at high tides.

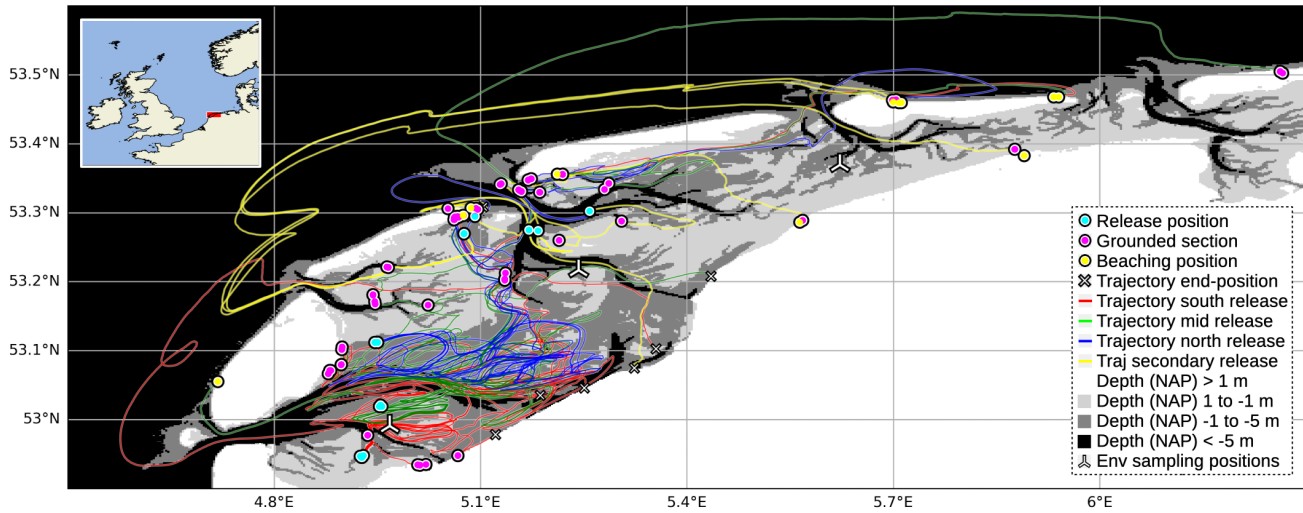

**Figure 2.** Trajectories (lines) and release/grounding locations (circles) of the campaign (van Sebille, 2024) drawn over a bathymetry map. This illustration synchronizes with Fig. 4 by matching the colors of the trajectories with the markings at the beginning of the release timeline, and by locating the three positions where environmental conditions (wind, water level) were sampled for reference. Maps and bathymetry generated with (see Met Office, 2010 - 2015; see Waddenregister)



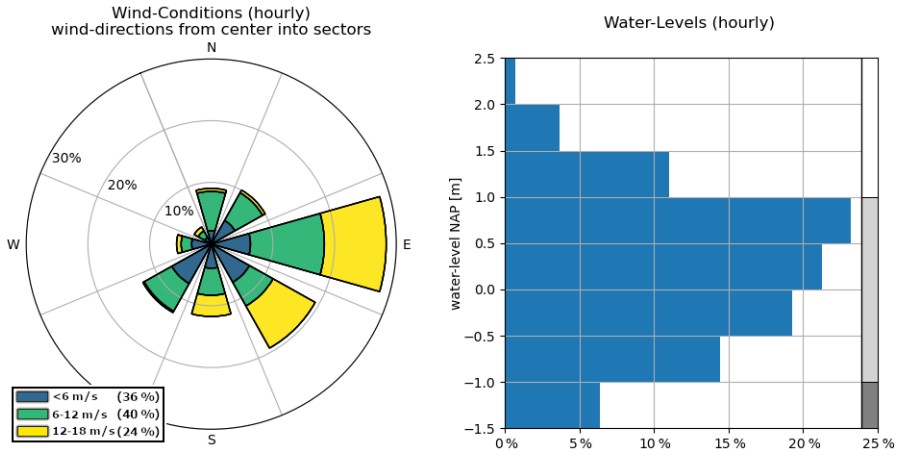

**Figure 3.** Histograms of hourly wind-conditions and water-levels from 14 November until 1 December 2023, at the central sampling position marked in Fig. 2. The gray-scale at the right border of the water level histogram is used in the bathymetry map in Fig. 2.

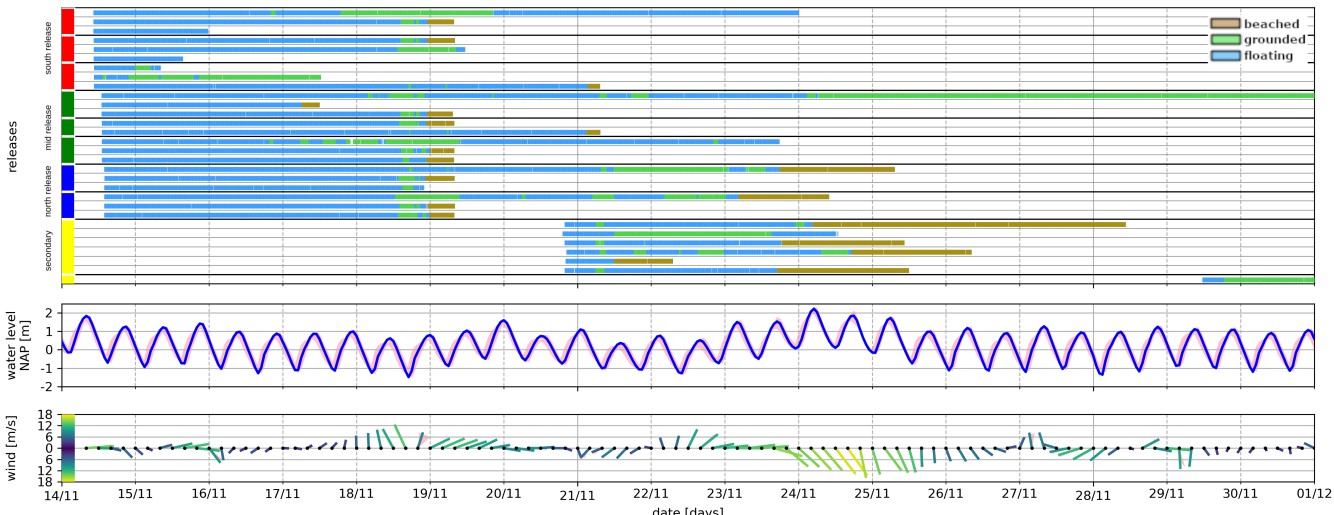

**Figure 4.** Timelines summarizing the campaign. The upper panel shows for each release the wet, dry and beached sections. The colors of the markings at the beginning of the release timeline are matched with the colors of the trajectories in Fig. 2, and white lines separate the grouped releases of the triangular pattern at each location. The middle and lower panels show the water level (hourly) and wind conditions (4h increments) for the same time period. Wind strength is illustrated by both color and length of each marker, and the wind direction is oriented from the central black dot towards the direction that the marker is pointing in. Both wind and water level are sampled at the three positions shown in Fig. 2, where the values of the central position are show in the foreground, and the values of the NW and SE positions define the bands of variation shown in the background.





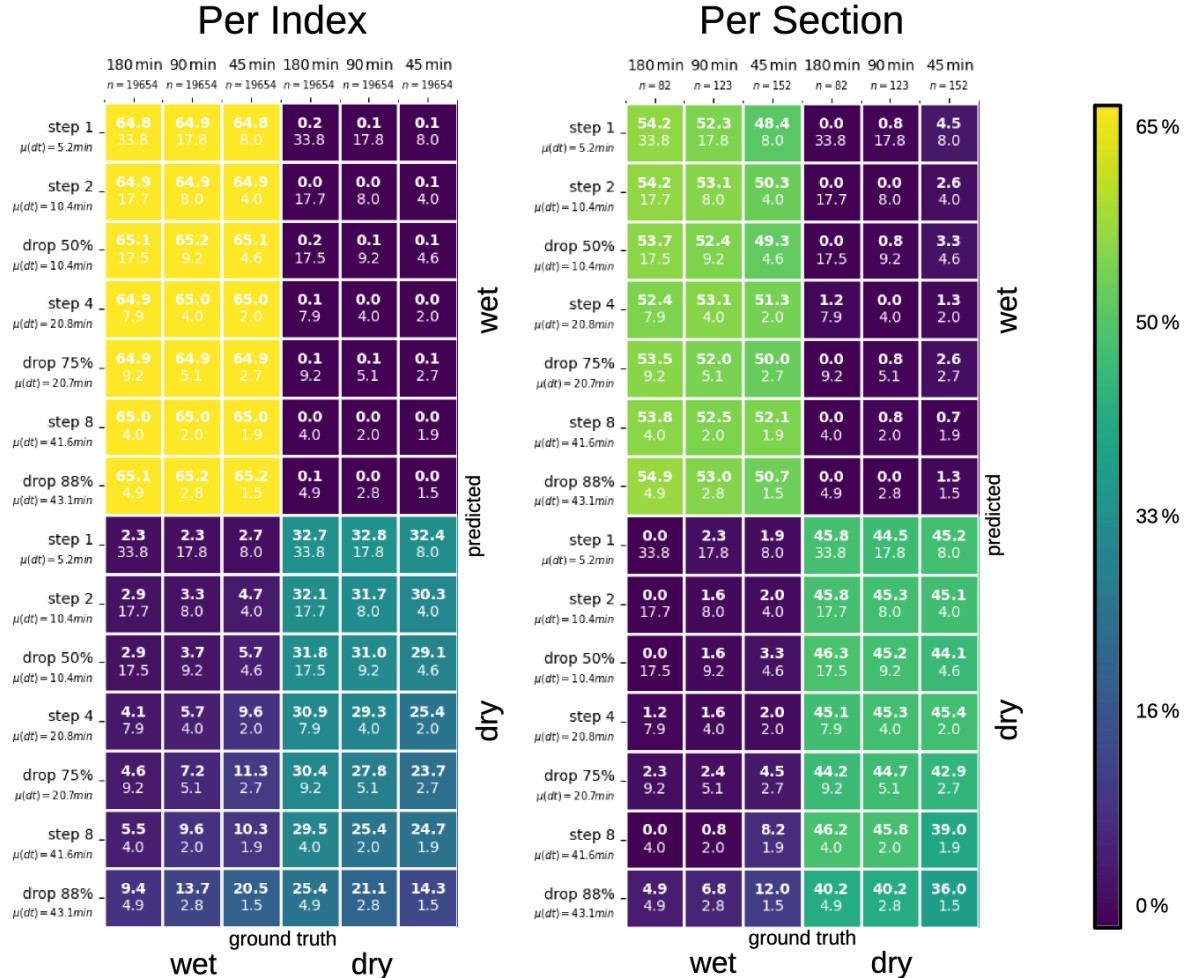

**Figure 5.** Collection of confusion matrices based on the two metrics and for different re-samplings of all drifter time-series. The horizontal direction lists the used time-windows of the grounding detection together with the total number of comparisons $n$, and the vertical direction lists the time-series together with the mean sampling interval $\mu(dt)$. In each square, the upper bold number shows the percentage of comparisons classified in the corresponding category, and the lower number shows the mean number of entries in the evaluated time-windows for the given time-series.



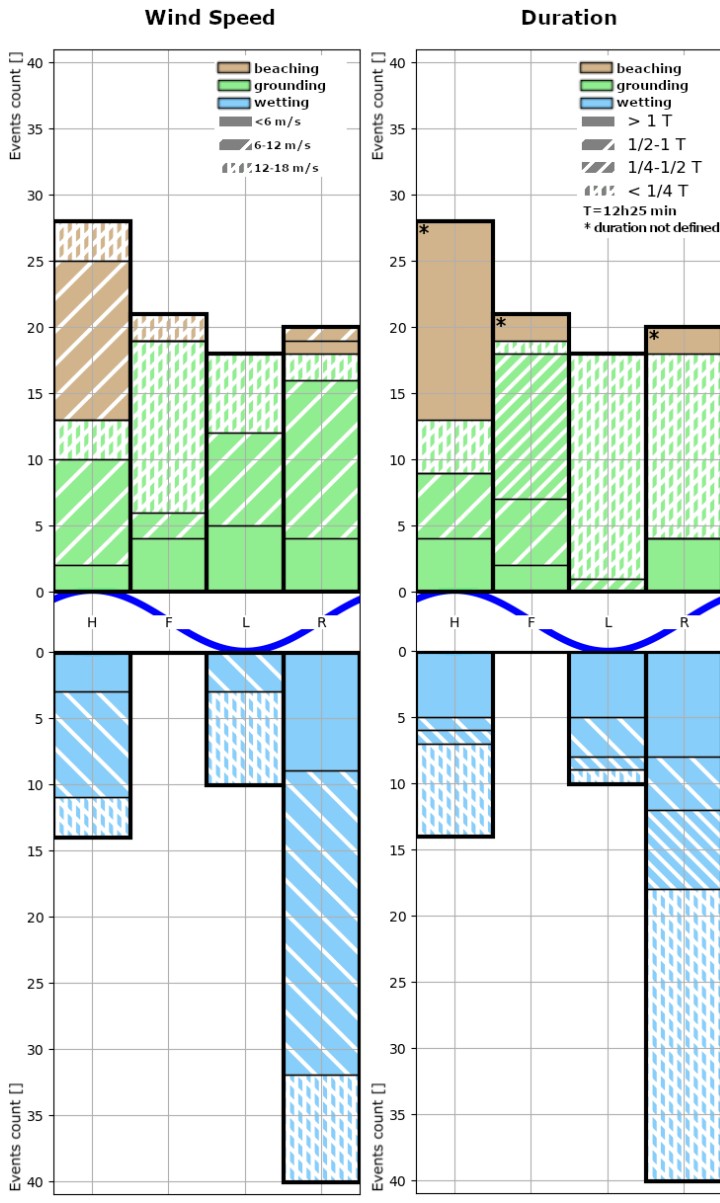

**Figure 6.** Grounding (upper half) and Wetting (lower half) occurrences with respect to the tidal phase (High, Falling, Low, Rising), separated by the event duration (left) or wind speed (right). Beaching events are stacked on top of drying events.



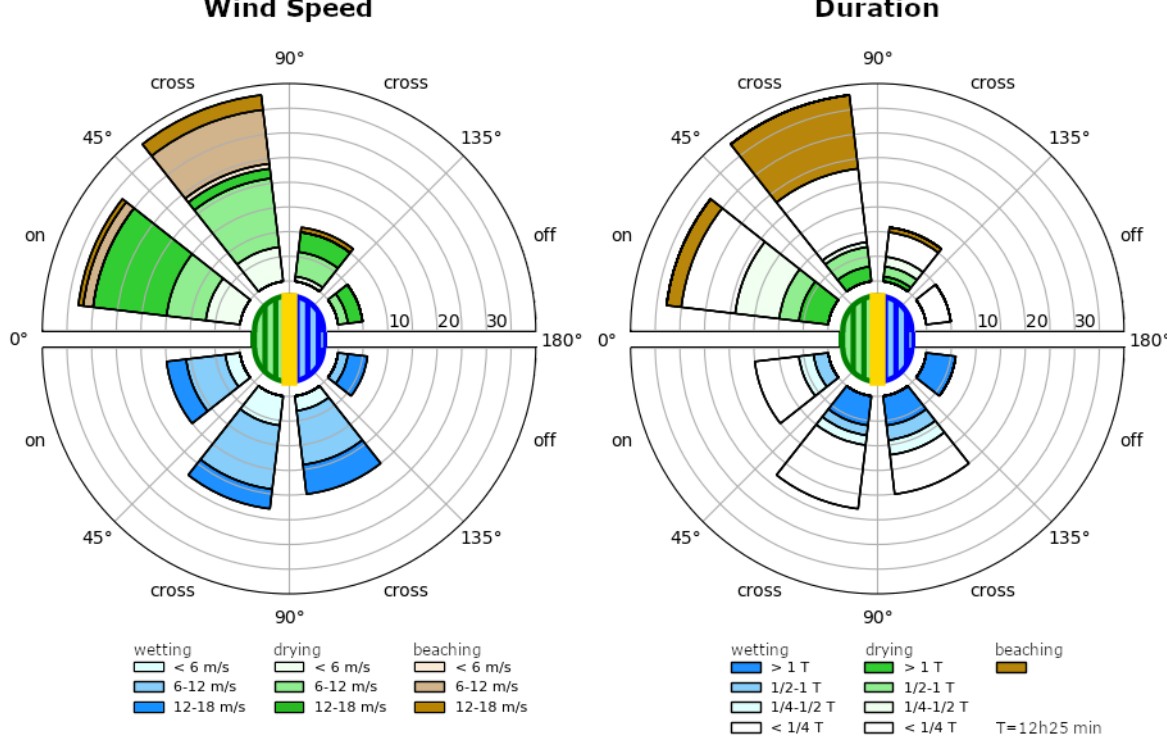

**Figure 7.** Grounding (upper hemisphere) and wetting (lower hemisphere) occurrences with respect to the wind direction along the local coast orientation, separated by wind speed (right) or the event duration (left). Beaching events are stacked on top of drying events. Land-coast-water orientation is illustrated in the center, and the wind orientation is indicated as the direction from the center towards each sector, as detailed in Table 3: The counts in each sector are including the events of the sector mirrored along the horizontal axis, giving rise to the notion of the four wind directions 'directly towards', 'diagonally towards', 'diagonally from' and 'directly from' shore.





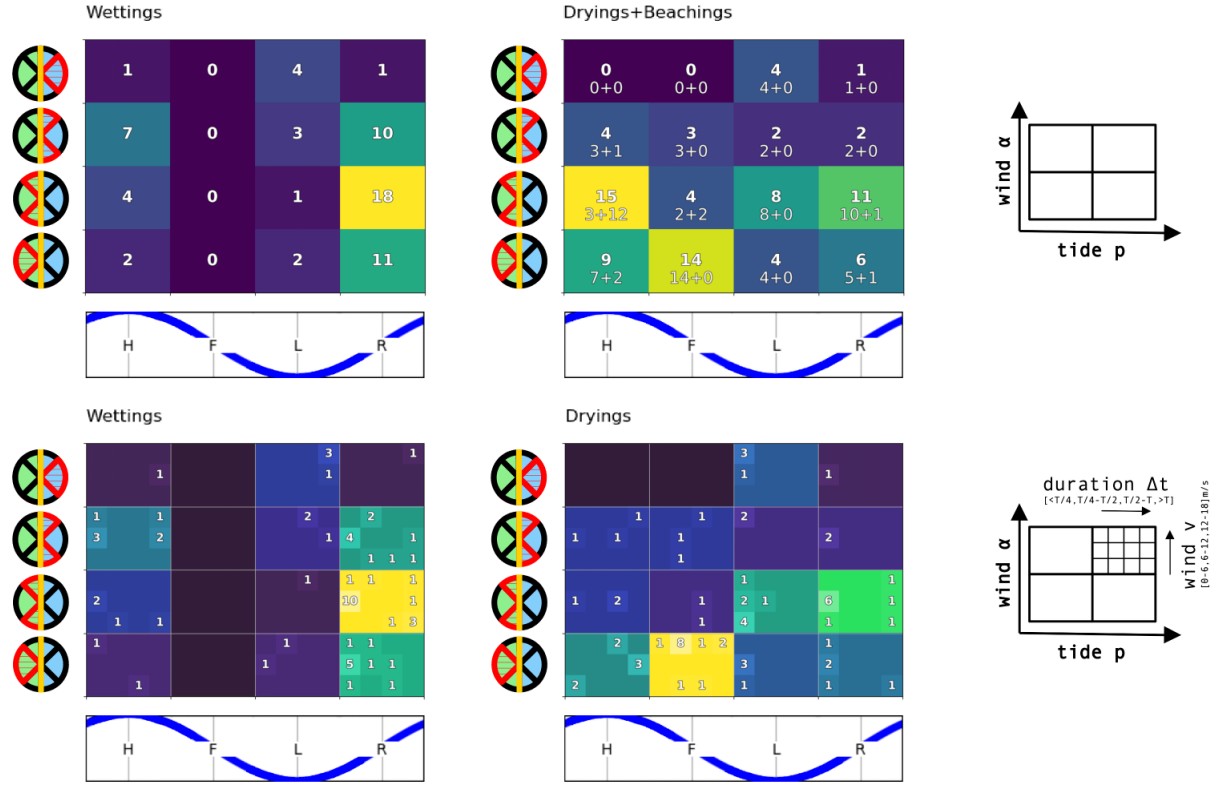

**Figure 8.** Wetting (left) and grounding (right) occurrences, separated by the environmental factors summarized in Table 3. The left column illustrates the internal organization of the plots: The upper row shows the counts of events per tidal phase and wind direction, as illustrated in the diagrams of the axes. Separate counts of drying and beaching events are written out as the sum below the total number. The lower row further details the counts of wettings and dryings by also separating them by wind speed and by section duration after each event.

# 4 Conclusions

We report a campaign in the Dutch Wadden Sea with 24 surface drifters where we identify 151 events related to groundings. In addition to manually coding the recorded trajectories, we also describe a statistical approach that separates sections of transportation from sections where the drifters are grounded. This statistic relies on four parameters that define the minimum duration of a detectable section, and the sensitivity to inaccuracies of the position measurements. While their values can be inferred from the sampling rate and measurement accuracy, physical arguments provide insights about which kind of phenomenon can be resolved. We evaluate the accuracy of the statistic and demonstrate that it is robust to changes in the average number of entries and irregular sampling.

We characterize the events and sections with four factors - the tidal phase, the wind direction and speed, and the duration of the sections - where we find that both wettings and groundings occur rarely during off-shore wind, wettings were not ob-




served during falling tides but often during rising tides, and while the number of groundings is much more uniformly spread over tidal cycles, the durations of grounded sections are shorter the closer the corresponding grounding happens before rising tides. We also observe that beachings occurred mostly during high tide and when winds were diagonally towards shore. These general observations agree with our intuition about grounding processes, which confirms the correctness of the analysis. The grounding characterization may be refined by including processes such as surges and neap-spring tidal cycles.

The detailed statistics provide a reference for future marine plastic studies and beach cleanup efforts, where models or weather forecasts can be used to estimate groundings and abundance of macroplastic at the coast. We find that mean to strong winds (6-18m/s) directed towards shore, and the time-window before and up to low tide favor the accumulation of grounded drifters. On the contrary, many wettings occur during cross shore wind, and before or up to high tide. These results are specific to the flat shape and relatively heavy weight of the Stokes Drifters. Putting this study into context with other objects, or deriving a corrective statistic to account for this specificity of the campaign can further improve the understanding of the beaching patterns of macroplastic.

A further limitation of the study is that our observations and results apply for the conditions of the campaign. The drifters were exposed to mostly south to east directed wind, and the occurring water levels and currents are heavily impacted by wind and semidiurnal tidal cycles. The complex topography and hydrodynamics of the Wadden Sea gives rise to multiple intertwined temporal and spatial scales. While this is an opportunity to study groundings under a large range of conditions, it also poses the challenge to manage the complexity in the data analysis.





*Code availability.* DOI https://doi.org/10.5281/zenodo.17865410 for GitHub repository 'mes-uu/Paper-Drifters-Groundings-Wadden-Sea-Floating-Macroplastic' and release 'Paper Submission Source-Code'. The computed figures are displayed in the online repository witinh the Jupyter Notebook files.

*Data availability.* The four used datasets of the drifter trajectories (van Sebille, 2024) bathymetry (see Waddenregister), water-levels (see Rijkswaterstaat), and wind-conditions (see ERA) are referenced in the GitHub repository of the code. Access to (see Rijkswaterstaat) requires authorization while the other datasets are open access. Manual codings of the grounding events are included in the GitHub repository.





**Appendix A: Description of the Manual Grounding Identification**

The drifter trajectories were inspected visually with plots that display the water level and bathymetry as shown in Fig. A1. We
define the following terms to describe the labeling of drifter trajectories:

– A **point** $p_i$ is a position in the scatter plot of the trajectory. The points are marked as small circles with a white interior
and black border.

– A **step** $s_i = (p_i, p_{i+1})$ consists of two successive points in the scatter plot, connected by a white line with a black border.

– A **trajectory** $(p_0, ..., p_N)$ is a succession of all position measurements for one drifter release

– The **direction of displacement** is the consistent direction of displacement during about three successive steps.

– The **jitter** is the random displacement that appears as star-like pattern of successive positions in the plot.

The wetting and drying events are then identified as described in Table A1.





**Table A1.** Identification criteria for grounding events.

| Category | Criteria | Implication of criteria |
|---|---|---|
| Grounded section | 1) A section of the scatter plot with jitter<br><br>2) The jitter is not directional, there is no slow drift over time<br><br>3) The event is at least 6 time-steps long | 1, 2) The position track is dominated by the GNSS accuracy and there is no clear positional trend in the data that indicated a displacement<br><br>3) This corresponds to a time interval of $6 \cdot 5$min = 30min |
| Drying point | 1) The latter point $p_{i+1}$ of the last step $s_i$ that continues the direction of the displacement before a grounded section.<br><br>2) The initial point $p_i$ of this step $s_i$ appears in the plot as fully separated from the overlapping positions in the grounded section and the latter point $p_{i+1}$ is the first point that overlaps. | 1,2) The last displacement before grounding is around 20m and follows the direction of displacement. In the grounded sections, the positions vary by a few meters, causing the plotted points to overlap.<br><br>A displacement of 20m is often observed before grounding and it is clearly above the GNSS accuracy |
| Wetting point | Opposite of the Drying point<br><br>1) The initial point pi of the first step $s_i$ that is disconnected from the overlapping points from the grounding jitter<br><br>2) The direction of the step aligns with the direction of displacement after the grounded section | 1,2) Parallel to drying points |



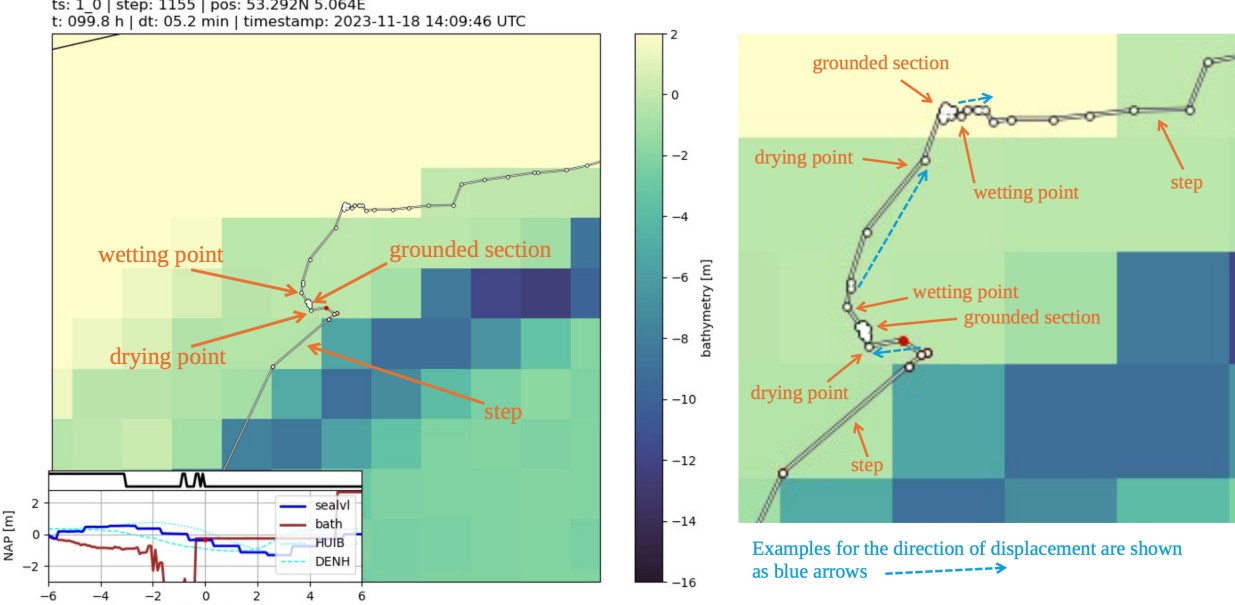

**Figure A1.** Trajectory plots for manual labeling with added annotations in orange (left) and a zoom-in of the same trajectory with more detailed annotations (right). Maps and bathymetry generated with (see Met Office, 2010 - 2015; see Waddenregister)



## Appendix B: Standardized Directional Change Computation

The recorded changes in position of GNSS receivers at rest are frequently at the resolution scale of the data, such that positions

often only change by a fixed minimal value increment along the North or East direction. An example of this can be seen in the insert of Fig. 1. This happens sporadically and creates a substantial number of directional changes that fall precisely on multiples of 90 degrees. We assume that this behavior varies for different receiver and logging solutions, and that asserting uniformly scattered directional changes is more universal. For this reason, we compute the directional changes $\alpha(t_i)$ from the recorded positions augmented with unbiased normal noise at a noise-level of the standard deviation of the GNSS receiver

accuracy $\sigma_{\text{gnss}}$

$$\alpha(t_i) := \angle(p(t_{i+1}) - p(t_i) + \epsilon(t_{i+1}), p(t_i) - p(t_{i-1}) + \epsilon(t_i)),$$

$$\text{with } \epsilon \sim \mathcal{N}([0,0], \mathbb{1}_{\text{2x2}} \cdot \sigma_{\text{gnss}}^2) .$$

This increases the standard deviation of a position recorded at rest by a factor of 1.4, which in our case lies below the scales that we rely on for the data analysis. Furthermore, this noise is only applied for the computation of the changes in directions.




## Appendix C: Sensitivity Analysis for the Local Wind Direction

The local wind direction at grounding sites is determined by computing the coastline orientation from the bathymetry map (see Waddenregister) and determining the wind direction with respect to this orientation (see ERA). The coastline orientation is computed by fitting a plane to the bathymetry values at the grounding location that lie within a given radius. The orientation of the plane then determines the coast orientation. The bathymetry map of the Wadden Sea resolves objects on multiple length-scales, and the dynamics of the grounding events are impacted by water movement that also involves multiple length-scales.

It is thus difficult to select a radius for the computation. For this reason, Fig. D1 contains a sensitivity analysis with choices of 0.5km and 2km for the radius, instead of the 1km used for the main analysis of Fig. 8. The lower boundary of 0.5km still includes multiple values of the bathymetry map that has a resolution of 0.1km, and the upper boundary of 2km coincides with larger topographic features like the minimum width of some of the islands.

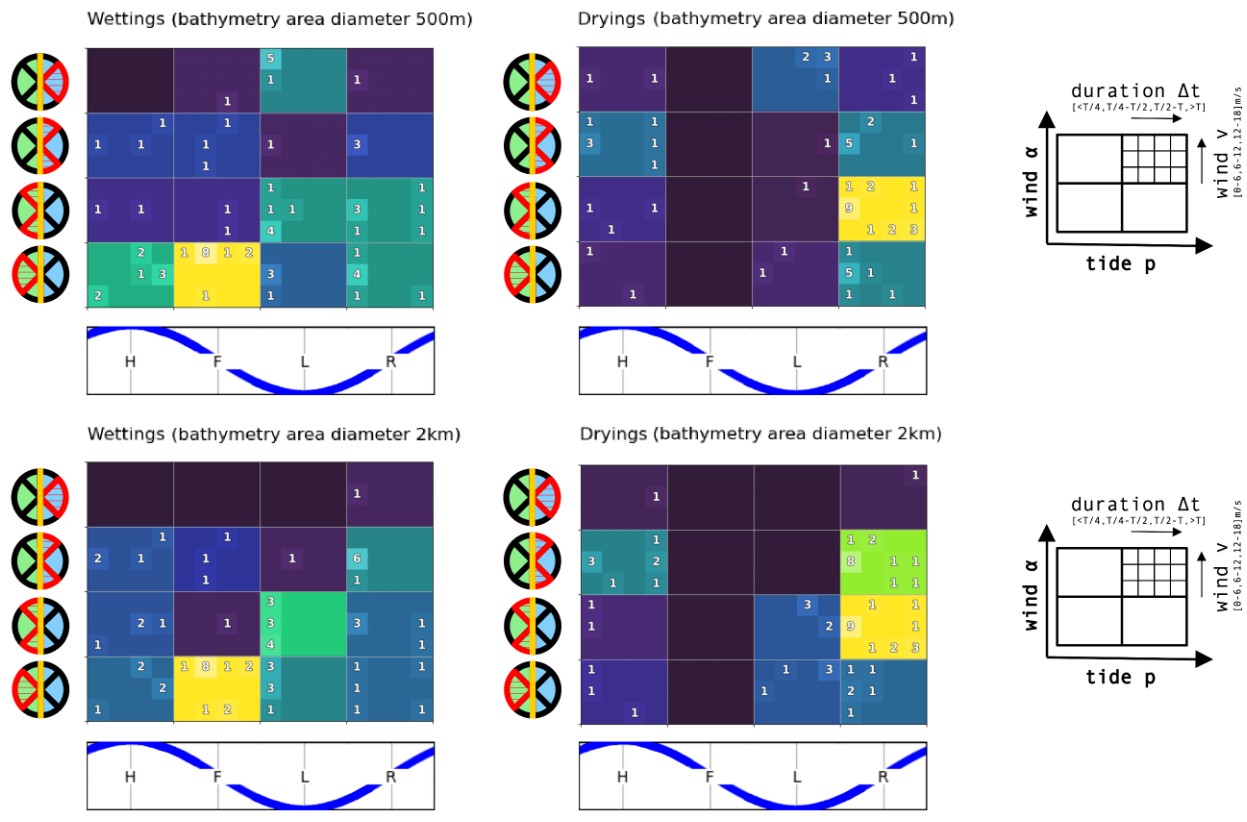

**Figure D1.** Sensitivity analysis for the computation of the local coastline orientation at the grounding sites. The plots here show the results for a radius of 0.5km and 2km, whereas the radius used in Fig. 8 of the main analysis is 1km.



## Appendix E: Drifter Sticker

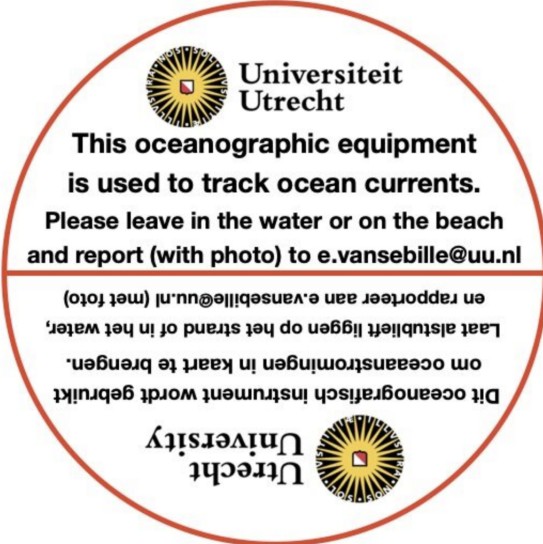

**Figure F1.** Sticker placed on both sides of the Stokes drifters.

*Author contributions.* MS constructed and implemented the study and data-analysis with input and feedback from EvS and RH

*Competing interests.* The authors declare that they have no conflict of interest.

*Acknowledgements.* This publication is part of the project "Tracing Marine Macroplastics by Unraveling the Ocean's Multiscale Transport Processes" with file number VI.C.222.025 of the research programme Vici ENW. We would like to thank Rühs S., Denes M.C., Altena B., Van Assenbergh M.G.R., Bertoncelj V., Stevens E.D., Brinkman F., and the Royal Netherlands Institute for Sea Research (NIOZ), in particular
Skipper Boon WJ., for participating in the drifter release at Texel. We would also like to thank the donors to the crowdfunding campaign of the university fund of Utrecht University, and the many citizens that reported and collected the drifters. We thank Jimena Medina-Rubio for manually coding the groundings in observations to validate the manual grounding identification method, and we thank Stuart G. Pearson and Jelle Soons for their input and feedback as experts on the Wadden Sea area.



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
