# Peer review of "Groundings of Drifters in the Wadden Sea Inform the Transport of Floating Macroplastic"

_EGUsphere, 2025_

## Referee Comment (RC1)

Review

Grounding of Drifters in the Wadden Sea Inform the Transport of

Floating Macroplastic

by Schneiter, Hut and van Sebille

**General comments**

The manuscript presents results of a measurement campaign where 30 drifters were released in the Dutch Wadden Sea (24 in a first release and six in a second release). The main goal of the paper is to first to identify the interaction events of the drifters with the coast and mudflats (groundings and wettings), and second, to relate these events to the coinciding wind characteristics and the tidal phase. 150 grounding events were identified. Due to the complex geometry of the system, the different locations of drifters, and the differences in the forcing (wind and tidal) conditions, it is not possible to perfectly link the grounding events to specific forcing conditions, but clear trends in the data are identified.

The paper has a concise and clear introduction. Then, the methods are described. These include the campaign, the drifters, and finally, the data collection and processing. The results give first a general description of the trajectories and the occurrence of grounding events. Then, they present a comparison of the methods to identify the grounding and wetting events. Finally, the results show under which conditions (tidal phase and wind velocity) the events occur.

In general, I find the results to be interesting and the campaign to be very important for the region, since it is the first Lagrangian field study. Although this paper is restricted to the analysis of grounding events motivated by the transport of macroplastic, this work can later complement recent Lagrangian numerical studies and earlier field and numerical work that related the transport to the forcing. However, I find that there is no attempt to put the results in context by discussing them in relation to what is already known about the system.

**Specific comments**

Although the results and the paper are mostly clear, I find certain passages to be difficult to follow because the authors obviate certain things, that are needed to fully grasp the methodology and the results. Hence, the paper would benefit from explicitly mentioning these things (as described in detail below).

l. 22 The formation of freshwater plumes is not the only effect of river runoffs. There is, for example, the estuarine circulation. In general, this sentence is a bit strange because to all the effects of the forcing mechanisms are mentioned.

Figure 1. Why are there steps in the water level? Notice also that "water level" are two words.

L. 153. This is, in general, not the correct way to compute the average of an angle, but I understand that it works here because it is the average of the absolute value and $-180° \leq \alpha \leq 180°$. This should be more explicitly mentioned here.

L. 225. It is unclear how do you "extract the tidal phase". Is it based on extracting only the M2 constituent or do you consider the maxima?

L. 226 -231 and Appendix C. The discussion about coastlines and shores is unclear. Sometimes, the drifters ground in the middle or edge of a mudflat. Are mudflats considered coastlines and shores? It seems to me that the wind direction is computed with respect to the local gradient in the bathymetry. If this is the case, it would seem clearer to define it in this way.

Figure 2. Consider splitting it into different subfigures for the different releases. It is currently difficult to follow the paths of the different releases. For example, I cannot distinguish if the events (grounding or beaching) are associated to green, red, yellow or blue lines when several come close together.

Lines 235-245. The discussion of Fig. 4 is rather limited, while there are several interesting points that can be highlighted and used later for the interpretation of the results. For example, the beaching of most drifters from the first release coincides with strong southerly winds, while the beaching of the drifters from the second release coincides with strong northwesterly winds. If one could easily follow the trajectories in Fig. 2, it might be possible to also observe a pattern on the locations of the beachings. This would highlight the importance of different wind directions and help motivate some of the later choices. Furthermore, these events of strong winds play a crucial role, as observed again in Fig. 7.

Fig. 3+Fig. 4. I am confused when I see the wind rose in Fig. 3 and the time series of wind in Fig. 4. First, it seems that the authors are not using the convection for the wind rose that it indicates the origin of the wind. Even after fixing this, I am having a difficult time seeing that the statistics of the time series are reflected in the wind rose.

Fig. 4. There is no discussion about the drifter that has the longest signal and gets grounded on 24/11 and remains grounded until December. Does it start floating eventually, and that is the reason why it is not considered as beached? What are the implications for understanding the results for the other drifters?

Fig. 5 + section 3.2. The explanation and interpretation of Fig. 5 is too limited. Are values of 65% or 45% acceptable? Good or bad? Or what does this mean for the use of the grounding statistic and the other results of the paper?

Conclusions. There is little attempt to explain the physical mechanisms behind what is observed or to relate the results to what is already known about the system. Furthermore, I disagree with the statement "The drifters were exposed to mostly south to east directed winds".

**Technical corrections**

L. 64. The first figure that is mentioned is Fig. 2.

L. 64. I suggest moving the information of the redeployment of six drifter to the beginning of the paragraph. I was first confused when looking at Fig. 2 and seeing eight release locations.

L. 80. Mention explicitly that GNSS gives the 3D location of the drifter.

L. 96. The part of the sentence "... removed excess ... by people on land" is unclear.

L. 102. Add a comma after "In the second step".

L. 112. Remove one "z" from "summarized".

Table 1. It might be good to explicitly mention that "drying" and "beaching" are two subcategories of grounding. As they are presented now, it seems at first that wetting, grounding, drying and beaching are four different things.

Figure 1. "water depth seabed level" is confusing. Is it the water depth? Or is it the seabed level with respect to NAP?

Figure 1, caption. Add ", respectively" after "wetting events".

L. 132. Do you mean the "discrete time derivative"?

Section 2.2.3. The adjectives to qualify certain quantities sound strange: the distance and the change in direction should be "small" not "low".

L. 140. Change "on water surfaces" to "on the water surface" and "velocities" to "speeds".